# Influence of Woven Glass-Fibre Prepreg Orientation on the Flexural Properties of a Sustainable Composite Honeycomb Sandwich Panel for Structural Applications

**DOI:** 10.3390/ma16145021

**Published:** 2023-07-15

**Authors:** Abd Latif Amir, Mohammad Ridzwan Ishak, Noorfaizal Yidris, Mohamed Yusoff Mohd Zuhri, Muhammad Rizal Muhammad Asyraf, Sharifah Zarina Syed Zakaria

**Affiliations:** 1Department of Aerospace Engineering, University Putra Malaysia (UPM), Serdang 43400, Selangor, Malaysia; nyidris@upm.edu.my; 2Aerospace Malaysia Research Centre (AMRC), University Putra Malaysia (UPM), Serdang 43400, Selangor, Malaysia; 3Laboratory of Biocomposite Technology, Institute of Tropical Forestry and Forest Products (INTROP), University Putra Malaysia (UPM), Serdang 43400, Selangor, Malaysia; 4Research Centre for Advanced Engineering Materials and Composites (AEMC), Department of Mechanical and Manufacturing Engineering, University Putra Malaysia (UPM), Serdang 43400, Selangor, Malaysia; zuhri@upm.edu.my; 5Engineering Design Research Group (EDRG), Faculty of Mechanical Engineering, University Teknologi Malaysia, Johor Bahru 81310, Johor, Malaysia; muhammadasyraf.mr@utm.my; 6Centre for Advanced Composite Materials (CACM), University Teknologi Malaysia, Johor Bahru 81310, Johor, Malaysia; 7Institute for Environment and Development (LESTARI), University Kebangsaan Malaysia (UKM), Bangi 43600, Selangor, Malaysia

**Keywords:** honeycomb sandwich, glass fibre, prepreg, orientation angle, cross-arm, sustainable structural material, transmission tower, flexural properties

## Abstract

Owing to the high potential application need in the aerospace and structural industry for honeycomb sandwich composite, the study on the flexural behaviour of sandwich composite structure has attracted attention in recent decades. The excellent bending behaviour of sandwich composite structures is based on their facesheet (FS) and core materials. This research studied the effect of woven glass-fibre prepreg orientation on the honeycomb sandwich panel. A three-point bending flexural test was done as per ASTM C393 standard by applying a 5 kN load on different orientation angles of woven glass-fibre prepreg honeycomb sandwich panel: α = 0°, 45° and 90°. The results show that most of the sandwich panel has almost the same failure mode during the three-point bending test. Additionally, the α = 0° orientation angle shows a higher maximum load prior to the first failure occurrence compared to others due to higher flexibility but lower stiffness. In addition, the woven glass-fibre prepreg orientation angle, α = 0°, has the maximum stress and flexural modulus, which directly depend upon the maximum load value obtained during the flexural test. In addition, the experimental results and analytical prediction for honeycomb sandwich deflection show good agreement. According to the result obtained, it is revealed that woven glass-fibre honeycomb sandwich panels with an α = 0° orientation is a good alternative compared to 45° and 90°, especially when better bending application is the main purpose. The final result of this research can be applied to enhance the properties of glass-fibre-reinforced polymer composite (GFRPC) cross-arm and enhance the existing cross-arm used in high transmission towers.

## 1. Introduction

A sandwich composite structure can be defined as a subset of a multi-layered structure consisting of outer facings with a core between them. Numerous alternative core designs have been employed, including balsa wood, foam, honeycomb, tetrahedral truss, corrugated core and various bioinspired cores [1,2,3,4,5,6]. The facing or facesheets are generally thinner as compared to the core, which allows the material to be strong and stiff with a lightweight property. Glass or carbon fibre is mostly used as a facesheet for sandwich composites structure as it provides a cost performance advantage and also offers superior resistance to environmental attack. At the same time, the facings can consist of a single metallic layer, or laminated or woven composite materials, which are also broadly employed to strengthen the sandwich structure [7,8,9,10]. The use of sandwich structure materials normally depends on the structure’s function and application, such as lifetime loading, availability and costing, where weight is critical, especially in aerospace, marine, automotive and lightweight structural applications [11,12,13]. Accordingly, a typical sandwich structure is somewhat similar to an I-beam where the flanges carry the largest share of bending and in-plane loads. Concurrently, the web sustains transverse shear, redistributes concentrated normal to the surface forces and maintains the structure’s integrity.

This research concentrates on the latest work of sandwich structures and their properties. To have a deep understanding of sandwich structure strength, researchers have made attempts using various methodologies, which can be categorized as core/structural modifications and matrix modifications, to improve the mechanical performance of sandwich composites [14,15,16,17,18]. Core modifications include peel stopper, corrugated core, brazed and hybrid core. At the same time, matrix modification involves various combinations of reinforcements, nanomaterials and varying fibre orientation. In this research, shape memory alloy wires are embedded between glass-fibre-reinforced epoxy composite layers in fibre metal laminate facesheet to enhance the sandwich panel’s behaviour in bending load. Chai et al. and Hosur et al. studied the low-velocity impact of sandwich structures due to dynamic response [19,20]. The application of magnetorheological and electrorheological fluid in the features and methods of control sandwich structures was studied by Eshaghi et al. [21]. Additionally, several researchers studied sandwich structure failure analysis such as bending, buckling, deformation, vibration and delamination [22,23,24,25,26]. The researchers were interested not only in experimental analysis, but they also studied theories and numerical methods pertaining to sandwich structures subject to different conditions [27,28,29]. In recent years, the application of composite sandwich structure has been widely used, so an increasing number of scholars have engaged in research and the application scope has gradually expanded. 

The qualities of sandwich constructions are significantly influenced by choosing a suitable material for the facesheet [30]. The function of the facesheets in sandwich structures is to support the bending forces placed on the structure. One facesheet is compressed, while the other is under tensile pressure during bending. When compressed, the thin facesheets cause local instability that may include the occurrence of buckling or debonding failure [31]. Several studies have examined facesheet materials, such as synthetic composites, reinforced petroleum-based composites, biobased composites and hybrid composites [32,33,34,35,36,37]. On the other hand, several experimental works have been found in the literature on behaviour and characterisation of different fibre orientation layups of the facesheets. Kharratzadeh et al. studied the delamination behaviour of sandwich structures made of plain woven glass/epoxy resin composites due to the effect of the interface fibre angle [38]. The results of this sandwich structure made by the hand lay-up method with a 0°/0° interface showed that the change in angles of the delamination interface has a negligible effect on magnitudes of initiation and propagation fractures toughness, together with the length of the fracture process zone. The study of fibre loading and orientation on the tensile and impact strength of composite sandwich materials between the three fibre orientations (0°, 30° and 60°) showed an improved strength for high fibre composite with 0° orientation [39]. Several researchers have focused on the fibre layup orientation of the facesheet, as shown in Table 1.

Three-point bending tests are usually performed to find the flexural and shear rigidities of sandwich panels [40,41]. To understand the properties of sandwich structures, the flexural behaviour, such as the force required to bend the structure and the resistance to flexure or stiffness of a material, need to be studied. Several researches have examined the flexural behaviour of sandwich panels, such as the flexural properties of sandwich panels with fibre metal laminate facesheets as investigated by Khalili et al. [42]. This study proved that the shear strength properties of the sandwich core play an essential role in the design of sandwich structures subjected to flexural loading. Li et al. investigated the flexural creep behaviour and life prediction of the GFRP–balsa sandwich structure [3]. A static three-point bending test was conducted to obtain ultimate strength and then the creep tests were run for 3000 h. The results show that the GFRP–balsa sandwich structure is not particularly excellent; therefore, it is recommended to conduct a repeatability test under allowable test conditions. By using the three-point bending test, Shi et al. studied the flexural strength of carbon-fibre sandwich panels with different core types [43]. The result on that study shows, a honeycomb-filled orthogrid core sandwich with aramid-fibre interfacial toughening improved strength and energy absorption for thin-walled structures used in various fields. 

**Table 1 materials-16-05021-t001:** Previous researches on fibre orientation layup.

Research	Facesheet Orientation	Finding	Ref.
Multiscale deformation and failure mechanism of different fibre orientations angle using digital image correlation	Woven GFRP composite (0°, 45°, 90°)	High degree of local deformation for 45° off-axis specimens, while fibre-pullout is the major failure mode for 0° and 90° orientation of facesheet fibre	[44]
Theoretical and experimental approaches for flexural and compression properties	Twill weave carbon/epoxy prepreg (0°, 90°)	Higher flexural stiffness and higher average critical collapse load values for sandwich composite fabricated using a single-core compared to multilayer-core sandwich composites	[45]
Theoretical and experimental approaches for warpage properties of honeycomb sandwich structure	Unidirectional carbon-fibre prepreg (0°, 45°, 90°)	The symmetric [45/−45/core/−45/45] configuration reduced warpage by 76%. Fibre orientation in the panel configurations has a significant effect on warpage	[46]
Failure behaviour and energy absorption performance of aluminium honeycomb sandwich beams	Unidirectional E-glass fibre prepreg and adhesive film (0°, 45°, 90°)	The change in fibre-laying angle, the peak force and the load-carrying capacity have no obvious change and the energy absorption capacity is almost same	[47]
Full-field strain measurements of the textile deformation on different scale and deformation behaviour of woven textile	Uniaxial and yarn glass woven fabrics weave (0°, 45°, 90°)	Full-field strain measurements should be routinely used during shear and tension tests of textile reinforcements	[48]

Table 1 shows previous studies on fibre orientation layup; however, studies are lacking on the flexural behaviour of woven prepreg glass-fibre orientation. Therefore, this research investigate the effect of woven prepreg glass-fibre orientation facesheet of honeycomb sandwich composite panels on flexural behaviour by applied a three-point bending test. In line with previous studies, the final result of this research can be applied to reinforce pultruded glass-fibre-reinforced polymer composite (PGFRPC) cross-arms used in high transmission towers [49,50]. To our knowledge, limited studies have been conducted based on woven prepreg glass-fibre composite with different fibre orientation facesheets, especially concurrently with flexural behaviour. We expect this study to provide better insight into the significance of variable fibre orientation on glass-fibre honeycomb composite structures in structural applications. 

## 2. Materials and Methods

### 2.1. Material Preparation

The preparation of tested specimens can be categorised into two major parts. The first is to fabricate the facesheets and the second one is to fabricate the sandwich panel. The facesheets were made of woven prepreg E-glass fibre (plain weave) with a surface density of 200 g/m^2^ and a thickness of 0.2 mm, provided from China. The prepreg adhesive of YPH-42T type with 18,000–25,000 cps viscosity and 1.21–1.25 g/m^2^ density was used. For this research, the E-glass fibre was used due to its higher strength and electrical resistivity [51]. This woven prepreg has axial stiffness, shear stiffness and bending strength of 18.5 Gpa, 2.47 Gpa and 100–110 Mpa, respectively. In this research, each facesheet was manufactured using the hand lay-up method with four layers of woven prepreg glass fibre stacked together at different fibre orientation angles for each facesheet. A schematic view of the orientation of the test specimens is illustrated in Figure 1.

The size of facesheet is followed by the size of the honeycomb core, which is 300 mm × 85 mm. The honeycomb core was made from A5052 aluminium grade, cell size of 3.4 mm and thickness of 25 mm. After the 4-layer woven prepreg glass-fibre facesheet was laid-up, the epoxy prepreg film was applied on both sides of the core to join the facesheets and the honeycomb core. The prepreg sample with four layers has shown higher ultimate force and pressure compared to non-prepreg glass laminates, as discussed by Ashraf et al. [52]. Next, pressure was applied with heat to join the facesheet and honeycomb core in a hot press machine for the curing process. The press machine temperature was set at 130 °C for 3 h [53], while the rising temperature rate was about 3 °C/min and the pressure was applied consistently at 5 bar by the hydraulic pump [52]. After 3 h, the press machine was turned off and the machine temperature was allowed to drop to 40 °C; at the same time, the specimens were maintained under the same pressure. Figure 2 shows the material used to fabricate the honeycomb sandwich panel and the hot press machine used in this research.

### 2.2. Experimental Setup

In line with other published works, the test specimens for the flexural test were prepared according to ASTM C 393 [54], and all the specimens were cut rectangular in cross sections with 200 mm length and 75 mm width, following the standard size of the testing sample [53]. For the top and bottom facesheet of each stacked honeycomb sandwich structure, four plies of woven glass-fibre prepreg were used. The thickness of the stacked honeycomb sandwich structure that is cured in a compression moulding system at constant curing temperature and pressure was around 28 mm. The specimens were fabricated and divided into three groups at different woven glass-fibre facesheet orientation angles, α = 0°, 45° and 90°. Figure 3 shows the flow chart of the overall experimental setup done in this research.

For each group, seven samples were tested and the average value was determined to provide valid results for the flexural properties of the honeycomb sandwich structure. Three-point flexural quasi-static tests were performed by using a computer-controlled electronic universal testing machine, INSTRON 3382. All these structural honeycombs were tested at a constant speed of 6 mm/min. The diameter of the support and loading steel cylinder used in this research was 25 mm, while the support span length was 150 mm, as shown in Figure 4. The temperature during the testing was 24 °C and the surrounding humidity was 53%.

### 2.3. Flexural Properties Based on ASTM Standard

In this study, the load–midspan deflection curve described the relationship between the mechanical response and structural flexural properties under a three-point bending test (3PB test). The values for flexural properties obtained from the average of seven repetitions in a quasi-static three-point bending test using the INSTRON universal testing machine were recorded, such as the maximum load, stress and flexural modulus. Three typical results for different angular orientations of the woven glass-fibre facesheet included α = 0°, 45° and 90°. This research used a 150 mm span length, as discussed in Section 2.2. Figure 5 illustrates the honeycomb sandwich panel dimension used in this research, and the average values were used in the calculation.

Afterward, the properties of the flexural behaviour, such as facesheet bending stress, core shear ultimate stress, transverse shear rigidity and core shear modulus of the honeycomb sandwich panels, were calculated using the load–deflection curve, as shown in Equation (1), following the ASTM C393/C393M and ASTM D7250/D7250M international standards [54,55]: (1)$$F_{s}=\frac{P_{max}}{\left(h+c\right)b}$$
where $F_{s}$ is the core shear ultimate stress (Mpa), $P_{max}$ is the maximum force prior to the first failure occurrence (N), h is the sandwich thickness (mm), c is the core thickness (mm) and b is the sandwich width (mm). The facing stress was calculated as:(2)$$\sigma =\frac{P_{max}S}{2t\left(h+c\right)b}$$
where σ is the facesheet bending stress (Mpa), *t* is the facesheet thickness (mm) and *S* is the span length (mm). The transverse shear rigidity was calculated by:(3)$$U=\frac{P(S-L)}{4\left[y-\frac{P(2S^{3}-3SL^{2}+L^{3})}{96D}\right]}$$
where *U* is the transverse shear rigidity (N), *L* is the load span length (mm), *y* is the midspan deflection (mm) and *D* is the flexural stiffness (N-mm^2^). In this research, *L* is equal to zero for the three-point bending condition due load span at the centre of the specimens. However, there are two variables in Equation (3) that make this equation difficult to solve. Therefore, elastic modulus, obtained by the quasi-static test, was used to find the flexural stiffness by using Equation (4), as shown below:(4)$$D=\frac{E\left(h^{3}-c^{3}\right)b}{12}$$
where the change in load (*P*) and change in midspan deflection (*y*) were considered at the maximum force required for the first failure occurrence. Then, the values obtained were substituted into Equation (3) to find the transverse shear rigidity for all kinds of woven-angle orientations. Next, the core shear modulus (*G*) was defined as:(5)$$G=\frac{U(d-2t)}{{\left(d-t\right)}^{2}b}$$

### 2.4. Analytical Prediction of Sandwich Beam Deflection

The total deflection (*δ*) of a honeycomb sandwich beam at the midpoint during the three-point bending is a sum of deflections due to bending of the facesheets and shear of the honeycomb core, referring to the theory of Allen [56]. The simple analytical models for the total deflection of a supported sandwich beam, referring to schematic diagrams in Figure 4 and Figure 5, can be given by:(6)$$\delta =\delta_{f}+\delta_{c}=\frac{PS^{3}}{48D}+\frac{PS}{4{\left(AG\right)}_{eq}}$$
where P is the force acting on the central point of the beam, *S* is the support span length, *A* is the cross sectional area of the sandwich beam, *D* is the flexural stiffness of the beam and *G* is the effective shear modulus of the core, while ${\left(AG\right)}_{eq}$ is the equivalent shear rigidity of honeycomb core [57,58]. Shear deformations for long beams with good shear stiffness are typically regarded as being minimal. However, shear deformation may be significant or even dominant in beams that are short and/or have relatively weak shear characteristics. It is noted that the expression for the bending response is based on the assumption of a small elastic deformation due to the small failure of the glass-fibre material. In addition, the geometrical nonlinearity has a negligible effect on the load–deflection response within the small deflection, according to research on the nonlinearity effects [59,60].

## 3. Results and Discussion

### 3.1. Failure Modes

Some failure and damage modes were observed during the test, either at the facesheet or the core, namely (a) indentation, (b) core shear, (c) core buckling, (d) delamination, (e) debonding and (f) fibre breakage/facesheet failure, as shown in Figure 6. Table 2 presents the failure modes of specimens for different facesheet orientation angles. As shown in the table, more than one failure mode was observed for each group of specimens by testing seven identical specimens. The loads corresponding to the failure modes are very close to each other, causing a change from one failure mode to another due to minor differences in geometry and location of the honeycomb cells.

Table 2 shows that most failure for the 0° and 90° orientations occur at the laminated facesheet instead of the core, compared to the 45° orientation. Additionally, the average peak load was higher for the 0° orientation followed by the 90° and 45° orientations. By using a three-point bending test, the load point due to local indentation based on core crushing under the indenter mostly occurred for the 90° orientation. This shows that the compressive stress for woven glass fibre at the 90° orientation exceeds the compressive strength of the aluminium honeycomb core as the peak load was achieved. Otherwise, the bending stiffness determines the degree of load spread out at the point of application. Results obtained from the flexural test showed that the indentation failure of the sandwich panel is mainly core dependent [61]. Core buckling was predicted and observed as the first damage mode in this research, especially for the 45° orientation. When the facesheet is much stiffer than the core, the core shear failure can occur due to the anisotropy of the honeycomb structure. Additionally, the shear strength of the core also depends on the loading direction [62].

Considering that the core is of sufficient stiffness, the global deflection of the sandwich tends to be large and, as a result, the high in-plane tensile forces will cause tensile cracking [63,64]. The highest tensile strength and elongation at breakage were obtained for the 0° orientation compared to the 45° and 90° orientation for the E-glass composite panel due to long and continuous glass fibres condition. In contrast, the other fibre orientation angles contained short, broken and discontinuous fibres as a result of the specimen cutting direction [65]. However, during the investigation of the fibre-orientation angle on the composite structure, the angle between the fibre axis and the loading direction must be considered. Furthermore, the delamination failure was also raised, as a standard damage initiation mode occurred in laminated composites, including the sandwich structure. Delamination between plies in laminates occurs when the interlaminar shear strength is exceeded and it is dependent on the peak load obtained. Additionally, the debonding mode failure between the facesheets and core occur due to weak bonding strength between them. Although not a catastrophic failure mode, debonding is also found to decrease the stiffness and strength significantly, and these were proven by Zhou et al. and Zhu et al. [66,67]. Due to the debonding failure mode occurring at the 45° orientation, most of the specimens failed at their core instead at a facesheet.

From a microscopic point of view, the reinforcing fibres prevent crack propagation by chemically bonding to the polymer matrix with covalent bonds [68]. The E-glass fibre composite used in this research can be classified as a continuous/aligned fibre-reinforced composite, which can offer anisotropy to various degrees, according to how many directions are involved and the number of fibres oriented along each direction [69]. E-glass fibre (“E” stands for electric), made from alumino-borosilicate with less than 1 wt% alkali oxides, was used in this research, compared to other types of glass fibre due to their great properties for electrical applications, since the final result of this research will be used to enhance the cross-arm in high transmission towers. Some reported advantages and disadvantages of different glass fibres are listed in Table 3. Recent researchers used the YPH-42T prepregs epoxy resin for the woven prepreg fibre because it is a solvent-free epoxy resin [70,71,72,73]. It is specifically designed to manufacture the high-strength carbon fibre, fibreglass and aramid fibreglass. It has high thermal deformation, impact resistance and excellent operational properties. It is suitable for manufacturing fishing rods, tennis rackets and other equipment requiring a low content of resin-moulding products. 

### 3.2. Load–Deflection Properties

The load midspan deflection curve was studied in this research, which presents the relationship between the mechanical responses and structural failure with different woven glass-fibre orientations under three-point bending tests. The midspan deflection was calculated based on averaging two placed at the midspan and on the bottom skin of each specimen. Three typical midspan load–deflection curves for glass-fibre orientation angles at α = 0°, 45° and 90° from the facesheet honeycomb sandwich panel are shown in Figure 7.

These curves were selected because they were the closest to the average energy absorption value in each corresponding group. Therefore, these curves presented typical performance for each type of sandwich panel. The load–deflection relation is shown as nonlinear in all the specimen sets, which was derived from the thermoplastic component.

Figure 7 shows two points were marked on the typical midspan deflection curve to explain the deflection behaviour for each different group of facesheet orientation. Up to point A, the aluminium honeycomb sandwich panel behaved as linear elastic. The linear elastic deformation for α = 45° was almost 1.6 times lower than α = 0° and 90° due to the occurrence of multiple fibre/matrix failures. During the loading, the applied load must be shared between the compliant matrix and the stiffer fibre bundles. This tendency of the fibres to reorient towards the loading direction in a woven composite as compared between α = 0° and 90° orientation [44]. Although the linear elastic behaviour for the facesheet orientation α = 90° was higher than that for α = 0°, the critical load achieved was lower compared to α = 0°. These observations prove that the honeycomb sandwich with α = 0° for the facesheet orientation has lower stiffness but higher flexibility compared to α = 90°.

The rigidity of sandwich panels typically decreases as the load moves from point A to point B. Point B represents the critical load, or the load that leads to the sandwich panel failure during the test. At this point, the loading area begins to experience localised indenting damage and core buckling failure. The sandwich panels’ capacity for supporting the loads increased until they reached the critical load at point B. Beyond point B, the ability to support loads declined and the large plastic deformation became apparent. In the best-case scenario, a biaxial symmetry response is anticipated for orthotropic materials like the one considered in this study. For instance, it is anticipated that the mechanical reaction of the woven glass fibre with a lay-up of both orientation angles, α = 0° and 90°, will be identical. However, in practice, the numbers of reinforcing fibres in the principal fibre angles are usually different, leading to relatively different mechanical and fracture properties [75,76]. The highest critical load reached in this research was 2620 N for α = 0°, followed by 2151 N for α = 90° and 1275 N for α = 45°.

### 3.3. Comparison between Analytical and Experimental Results

This section compares the analytical and experimental results for all types of woven glass-fibre facesheet orientations of honeycomb sandwich beam under a three-point bending load. The experiment was carried out to investigate the mechanical response of the sandwich beams. Table 4 lists the total deflection of the specimens and the bending stiffness, experimental results and analytical predictions. The total deflection of the specimens was calculated from Equation (6) and the bending deflection error was calculated as well. The average value from seven samples for each type of woven glass-fibre prepreg orientation was used in this calculation.

It was found that when the shear stiffness is invariant, the slope of load vs. midspan deflection is more prominent and the percent of shear deformation in the total deformation becomes more significant as the orientation of the glass fibre for the facesheet becomes α = 0°. When the shear stiffness is changed to be smaller, shear deformation can be more significant. However, Table 4 shows that the results of P/δ show good agreement with the predicted value. The main reason for the deviation between the analytical calculation and the experimental results is that the shear deflection was restricted by the end. Another reason is that the total shear deformation of both the facesheet and the core was considered in this study; therefore, the individual deformations are not considered. Additionally, the strength of the honeycomb sandwich panel with α = 0° for the glass-fibre woven prepreg orientation of the facesheet is much greater than for α = 90° and 45° due to the high critical load, which is the main criterion needed to enhance the pultruded glass-fibre reinforced polymer composite (PGFRPC) cross-arm beam.

### 3.4. Flexural Behaviour Characterisation

Table 5 shows the flexural behaviour obtained by the quasi-static three-point bending test for different orientation angles of the facesheet lay-up. It shows that the maximum load or force required prior to the first failure occurrence for orientation angle α = 0°, 90° and 45° was 2747.78, 2478.25 and 1454.91 N, respectively. The highest maximum load was occurred at the 0° of glass-fibre orientation facesheet, followed by 90° and the lowest at 45°. In ideal circumstances, a biaxial symmetry response is anticipated for orthotropic materials like the one considered in this work, meaning that the mechanical reaction of the specimens with angles = 0° and 90° is anticipated to be the same. However, during the testing, specimens derived from the two primary fibre orientations like 0° and 90°, the actual quantity of reinforcing fibre in the principal fibre angles was typically different, resulting in somewhat variable mechanical and fracture properties [77]. Therefore, a higher load is required to break the strong fibre bonding between the facesheet layers of the sandwich panels for 0° compared to 90° orientation angle. Additionally, a large shear deformation could occur for the fibre/matrix interface, especially at the 45° orientation angle during the quasi-static test, which can dominate void nucleation and coalescence within the fibre/matrix interface. As Godara and Raebe argued, this situation leads to multiple angle fracture planes in the laminated fibre [78]. They also stated that the applied load is resolved and induces shear flow in the matrix even at the lower applied load when the fibres are oriented at 45°. Hence, the honeycomb sandwich panel with the 45° orientation angle began to fail earlier with a lower load needed, compared to the 0° and 90° orientation angles.

A decrease in the facesheet bending strength for the sandwich panels was observed with the change in orientation angle of the facesheet lay-up: α = 0°, 90° and 45° were 27.22, 23.75 and 12.95 MPa, respectively (refer to Figure 8a). In the 3PB test, the reinforcing fibre, which has greater stiffness and less ductility, carries most of the load. Up until the fibre’s failure strain, the loading persists and is mostly supported by the fibre. When a fibre fails, the load is immediately transferred to the matrix, which causes the samples to completely fail. Similar to the core shear ultimate strength, the values also decrease as the orientation angle of facesheet lay-up changes: α = 0°, 90° and 45° were 0.70, 0.65 and 0.34 MPa, respectively (refer to Figure 8b). Indeed, compared to 45°, the core shear ultimate strength for 0° and 90° has a slight difference, suspected due to high energy absorption and strong fibre/matrix bonding.

However, the core shear modulus and transverse shear rigidity values show irregular variation compared to the facing bending strength and core shear ultimate strength for the three different orientation angles, α = 0°, 90° and 45°, where the core shear modulus values were 15.69, 29.55 and 9.25 MPa for α = 0°, 90° and 45° from the orientation angle of the facesheet lay-up, respectively (refer to Figure 8c). The transverse shear rigidity values were 30,993, 56,896 and 18,235 N, respectively, as shown in Figure 8d. It was found that both the core shear modulus and the transverse shear rigidity first increase by changing the orientation angle of the facesheet from α = 0° to α = 90°, and after that decrease for the angle α = 45°. A possible reason for this uneven trend in the transverse shear rigidity is due to the influence of both core geometry and core skin adhesion [79]; in this research, the prepreg adhesive was used to attach the core and facesheet of the honeycomb sandwich panel. Another possible reason for the irregular variation may be due to the core shear modulus dependence upon the peak load and cross-head displacement values. Despite an increase in the peak load values, by changing the orientation angle of the facesheet from α = 0° to α = 90°, the deflection values at respective loads were different, which define that the decreasing and increasing trend occurred. From this result, it can be stated that the transverse shear rigidity for α = 90° is considerably higher as compared to α = 0° and 45°.

## 4. Conclusions

The purpose of this study was to evaluate the different orientation angle behaviour of woven glass-fibre prepreg facesheet for honeycomb sandwich composite panels due to the three-point bending test. Based on the results obtained from experimental and numerical calculation, the following conclusions can be drawn:Most of the failure occurred at the facesheet instead of the core for the 0° and 90° orientations, due to the interlaminar shear strength exceeding their limit due to the load obtained. However, the core failure often occurs at the 45° orientation due to the weak bonding strength and decreased stiffness for this type of orientation.Although the linear elastic behaviour for the orientation angle α = 90° shows as higher than α = 0°, the maximum load achieved prior to the first failure occurrence for α = 0° is higher than that at α = 90°, which proved that the honeycomb sandwich with α = 0° for the facesheet orientation has lower stiffness but higher flexibility compared to α = 90°.The linear elastic deformation for α = 45° was almost 1.6 times lower than that for α = 0° and 90° due to multiple fibre/matrix failures that occurred.A decrease in the facesheet bending strength, core shear ultimate strength and core shear modulus of sandwich panels was observed, with the change in the orientation angle of the facesheet lay-up from α = 0°, 90° and 45°, correspondingly.However, the transverse shear rigidity values show an irregular variation due to the influence of both the core geometry and a core facesheet adhesion, where the prepreg adhesive was used in this study to attach the core and facesheet of the honeycomb sandwich panel. From this result, it was shown that the transverse shear rigidity for α = 90° is considerably higher compared to α = 0° and 45°.Several limitations that should be studied in the future are the type of woven prepreg used for the facesheet, the effect of honeycomb core thickness, the effect of facesheet ply lay-up, and the effect of span length, along with a better study of the failure mode for the large-scale panel. Additionally, scanning electron microscopy (SEM) measurement should be carried out to show the dispersion and morphology of the glass fibre in the composite.Moreover, the behaviour based on sandwich panels needs to be studied under long-term loading to consider the creep effect of the woven prepreg honeycomb sandwich base.

Considering these results and implications, this paper highlights the behaviour of the woven glass-fibre prepreg honeycomb sandwich panel and the possibility for further research in evaluating other mechanical properties. According to the results obtained, it is revealed that a woven glass-fibre honeycomb sandwich panel with α = 0° orientation is a good alternative as compared to 45° and 90°, especially when an improved bending application is the main purpose. The final result of this research can be applied to enhance the properties of the pultruded glass-fibre reinforced polymer composite (PGFRPC) cross-arm to substitute for the existing cross-arm used in high transmission towers.

## Figures and Tables

**Figure 1 materials-16-05021-f001:**
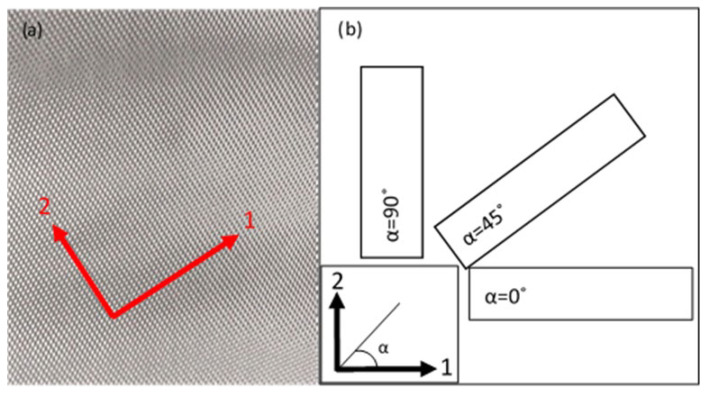
Schematic diagram of woven prepreg glass fibre used in this research. (**a**) Woven glass-fibre structure with the principal directions marked as 1 and 2. (**b**) Orientation of the facesheet fibre fabric extracted from the original sheet.

**Figure 2 materials-16-05021-f002:**
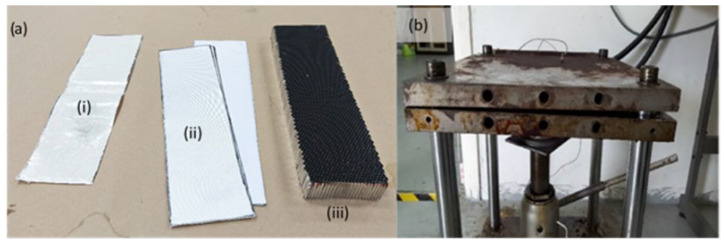
(**a**) Honeycomb sandwich panel fabrication material: (**i**) epoxy prepreg film; (**ii**) four layers of woven prepreg glass-fibre facesheet; and (**iii**) aluminium honeycomb core. (**b**) Hot press machine for the curing process.

**Figure 3 materials-16-05021-f003:**
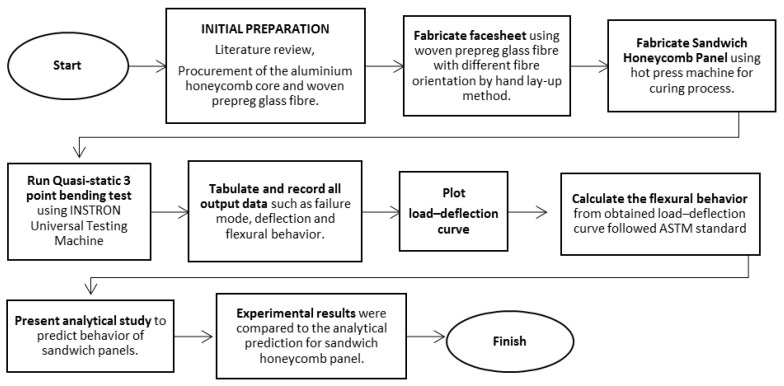
Flow chart of overall experimental setup.

**Figure 4 materials-16-05021-f004:**
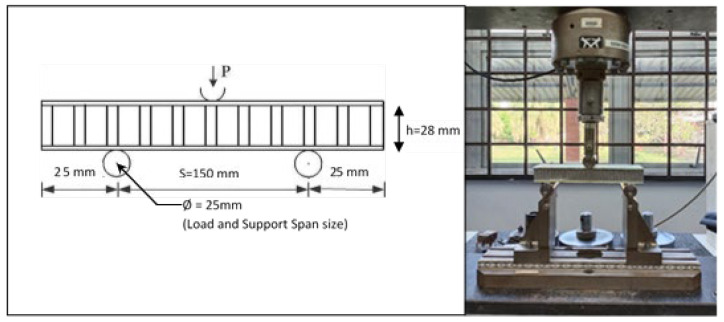
Honeycomb sandwich panel under quasi-static three-point bending test.

**Figure 5 materials-16-05021-f005:**
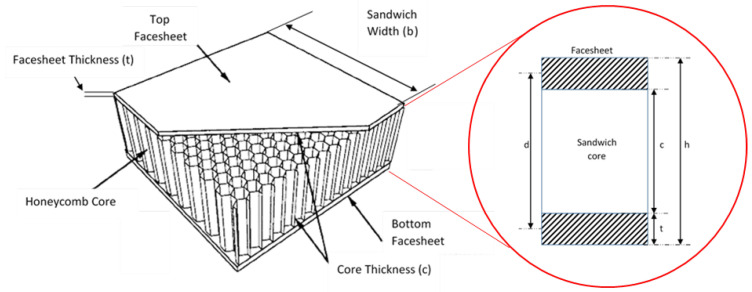
Dimensions of the honeycomb sandwich panel (d is distance between the centre line of upper and lower facesheet, while h is the total thickness of the sandwich honeycomb panel).

**Figure 6 materials-16-05021-f006:**
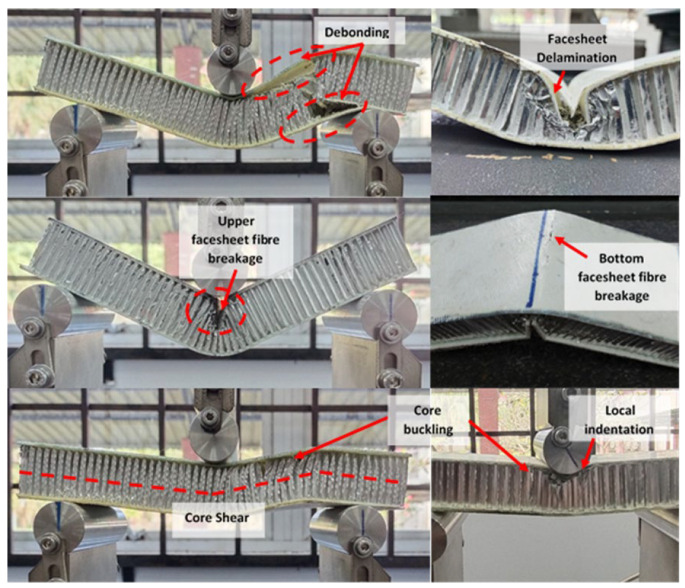
Failure modes of an aluminium honeycomb sandwich panel following a quasi-static three-point bending flexural test.

**Figure 7 materials-16-05021-f007:**
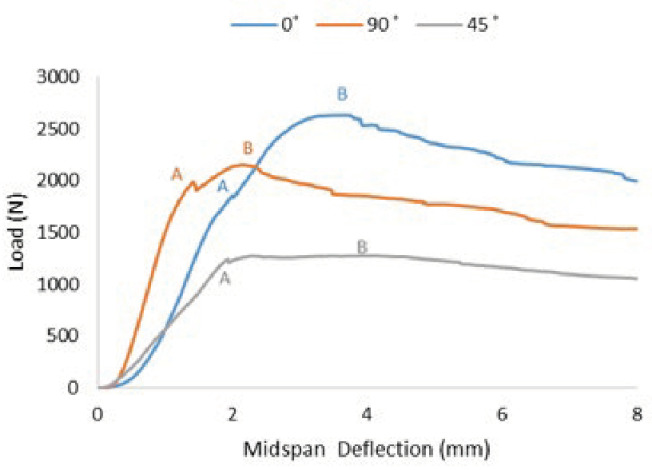
Load–deflection behaviour for different woven-fibre orientation for honeycomb sandwich structure where point A is the last point of linear elastic behaviour and point B is the maximum load required to fail the sandwich panel.

**Figure 8 materials-16-05021-f008:**
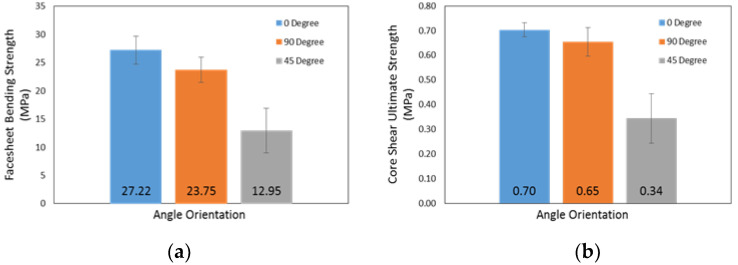
Properties of honeycomb sandwich panel for α = 0°, 90° and 45° orientation angle: (**a**) facesheet bending strength; (**b**) core shear ultimate strength; (**c**) core shear modulus; and (**d**) transverse shear rigidity.

**Table 2 materials-16-05021-t002:** Summary of failure modes for the three-point bending test.

Orientation Angle (°)	Peak Load (N)		* Failure Modes					
α	AVG	SD	DLM	IND	DBD	CS	CB	FF
0°	2747.79	206.64	5	3	2	2	2	5
90°	2475.80	490.38	2	5	2	1	3	5
45°	1352.50	1023.07	1	3	5	4	6	3

Note: AVG = average, SD = standard deviation, DLM = delamination, IND = indentation, DBD = debonding, CS = core shear and CB = core buckling, FF = facesheet failure. * Numbers of specimens out of seven identical specimens with identical failure mode for each orientation angle group.

**Table 3 materials-16-05021-t003:** Glass-fibre classification and application [69,74].

Type	Composition	Characteristics	Application
A-glass	Alkali-lime with little or no boron oxide	Not very resistant to alkali	When alkali resistance is not a requirement
C-glass (T-glass)	Alkali-lime with high boron oxide content	Resistant to chemical attack and most acids that dissolve E-glass	When higher chemical resistance to acid is required, for example, glass staple fibres
D-glass	Borosilicate	High dielectric constant	When high dielectric constant is preferred
E-glass	Alumino-borosilicate with less than 1 wt% alkali oxides	Not chloride-ion resistant; E-glass surface is soluble	Mainly for glass-reinforced plastics; originally for electrical applications
E-CR-glass	Alumino-lime silicate with less than 1 wt% alkali oxides	High acid resistance	When high acid resistance is required
R-glass	Alumino-silicate without MgO or CaO	Good mechanical properties	With high mechanical requirements
S-glass	Alumino-silicate without CaO but with high MgO content	Highest tensile strength among all types of fibre	Aircraft components and missile casings, when high tensile strength is required

**Table 4 materials-16-05021-t004:** Three-point bending deformation of honeycomb sandwich panels along with predicted and measured deflection.

Orientation Angle (°) α	Experimental, δ (mm)	Analytical, δ (mm)	Error % (δ)	Experimental, P/δ (N/mm)	Analytical, P/δ (N/mm)	Error % (P/δ)
0°	3.72	3.34	12.22	826.48	919.22	1.19
90°	1.83	1.95	26.80	1517.22	1368.17	2.12
45°	2.90	2.30	58.58	486.27	738.54	5.99

**Table 5 materials-16-05021-t005:** Flexural behaviour of the honeycomb sandwich panel obtained by the quasi-static 3PB test.

Orientation Angle (°) α	Max. Load (N) *P*	Max. Stress (MPa) *σ*	Flex. Moduli (MPa) *E*
0°	2747.78	10.60	1148.26
90°	2478.25	9.81	1440.20
45°	1454.91	5.56	706.16

## Data Availability

The data used to support the findings of this study are included within the article.

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
