# Peer review of "Influence of Woven Glass-Fibre Prepreg Orientation on the Flexural Properties of a Sustainable Composite Honeycomb Sandwich Panel for Structural Applications"

_materials, 2023, doi:10.3390/ma16145021_

Round 1

Reviewer 1 Report

1. The authors studied the four-layer facesheet structure. What is the effect of facesheet layers on the flexural property of composite materials? The work is interesting, but the experimental work can be improved. More experiments should be carried out.

 2. What is the effect of the thickness of the honeycomb sandwich core on the flexural property of composite materials?

 3. Line 35: a god agreement or a good agreement?

 4. Lines 125-126: … using hand lay-up method with method with were the woven prepreg glass fibre were stacked together …

This sentence is confusing. Rewrite it please.

 5. Line 127: … each facesheets

facesheets facesheet

 6. Lines 251-253: The load mid-span deflection curve was study in this research, which present the relationship between mechanical responses and structural failed with different woven glass fibre orientation under 3 point bending testing.

was study was studied

present presents

orientation orientations

The authors should carefully check the writings of the manuscript.

Author Response

Dear editor,

Thanks for your letter and the thoughtful comments from the referees about our paper entitled “Influence of Woven Glass-Fibre Prepreg Orientation on Flexural Properties of a Sustainable Composite Honeycomb Sandwich Panel for Structural Applications” by A. L. Amir, M. R. Ishak, N. Yidris, M.Y.M. Zuhri, M. R. M. Asyraf and S. Z. S. Zakaria for consideration to be published in Materials Journal. We carefully analysed all the comments and these comments are very valuable and helpful for perfecting and modifying our manuscript, and also have important guiding significance for our research. Therefore, we carefully checked the manuscript and revised it according to each comment. Consequently, we feel that our manuscript is substantially strengthened. Revised portion are marked using yellow background in the revised manuscript. The detailed corrections in the paper and the responses to the reviewer’s comments are as the following list of revisions.

We look forward to your positive response. If you have any question about this paper, please don’t hesitate to let us know. We hope these revisions will make it more acceptable for publication. Thank you.

Sincerely yours,

Amir

Faculty of Engineering,

Universiti Putra Malaysia,

81310, UPM Serdang,

Selangor, Malaysia

Reviewer 1

  1. The authors studied the four-layer facesheet structure. What is the effect of facesheet layers on the flexural property of composite materials? The work is interesting, but the experimental work can be improved. More experiments should be carried out.

Dear reviewer, thanks for the comment. In this study, only four-layer facesheet structure is considered based on the previous study by W.Ashraf et al. (as mention in line 151-154). The effect of facesheets layers on flexural property will become a gap for this paper to be fulfil in the future as discussed in conclusion as stated in line 452-457.

  1. What is the effect of the thickness of the honeycomb sandwich core on the flexural property of composite materials?

Dear reviewer, thanks a lot for the constructive comment. In this study, only the aluminum honeycomb core with 25mm thickness are considered. The effect of the thickness for the honeycomb sandwich core on the flexural property will become a gap for this paper to be fulfil in the future as discussed in conclusion.

  1. Line 35: a god agreement or a good agreement?

Dear reviewer, thanks a lot for the comment. The sentence has been corrected.

  1. Lines 125-126: … using hand lay-up method with method with were the woven prepreg glass fibre were stacked together …

This sentence is confusing. Rewrite it please.

Thanks a lot for your comment. The sentence has been rewrite. Thank you.

  1. Line 127: … each facesheets

facesheets → facesheet

Dear reviewer, thanks a lot for the comment. The sentence has been corrected.

  1. Lines 251-253: The load mid-span deflection curve was study in this research, which present the relationship between mechanical responses and structural failed with different woven glass fibre orientation under 3 point bending testing.

was study → was studied

present → presents

orientation → orientations

Dear reviewer, we have revised the manuscript as your comment. Thank you very much.

Reviewer 2 Report

The English expression might be improved by a native English speaker

Author Response

Dear editor,

Thanks for your letter and the thoughtful comments from the referees about our paper entitled “Influence of Woven Glass-Fibre Prepreg Orientation on Flexural Properties of a Sustainable Composite Honeycomb Sandwich Panel for Structural Applications” by A. L. Amir, M. R. Ishak, N. Yidris, M.Y.M. Zuhri, M. R. M. Asyraf and S. Z. S. Zakaria for consideration to be published in Materials Journal. We carefully analysed all the comments and these comments are very valuable and helpful for perfecting and modifying our manuscript, and also have important guiding significance for our research. Therefore, we carefully checked the manuscript and revised it according to each comment. Consequently, we feel that our manuscript is substantially strengthened. Revised portion are marked using yellow background in the revised manuscript. The detailed corrections in the paper and the responses to the reviewer’s comments are as the following list of revisions.

We look forward to your positive response. If you have any question about this paper, please don’t hesitate to let us know. We hope these revisions will make it more acceptable for publication. Thank you.

Sincerely yours,

Amir

Faculty of Engineering,

Universiti Putra Malaysia,

81310, UPM Serdang,

Selangor, Malaysia

Reviewer 2

  1. The title of this article is not a proper noun. Such as, “Glass Fibre Prepreg Honeycomb Sandwich Composite”. Please standardize this.

Dear reviewer, thanks for the constructive comment. The title of this article has been change to “Influence of Woven Glass-Fibre Prepreg Orientation on Flexural Properties of a Sustainable Composite Honeycomb Sandwich Panel for Structural Applications”. Thank you.

  1. In this manuscript, the sentence described in abstract, such as “In addition, the experimental results and analytical prediction for sandwich honeycomb deflection shows a god agreement”, has a word error.

Dear reviewer, thanks a lot for the comment. The sentence has been corrected.

  1. In the introduction, the significance of this study cannot be found. In this manuscript, the effect of woven prepreg glass fibre composite with different fibre orientation facesheets on flexural behavior of laminate, which is similar to that of layup angle on flexural behavior of laminate. The introduction needs to be improved.

Dear reviewer, we have revised the introduction in this manuscript as your comments. Thanks a lot.

  1. In Figure 6, the failure modes should be clearly marked in images. The subtitle should be added.

Dear reviewer, thanks a lot. The failure mode in Figure 6 has been marked clearly.

  1. The description about the experimental fracture mechanism of laminates subjected to compressive loading is little. More sentences should be added to exhibit the underlying mechanism.

Dear reviewer, thanks for the constructive comment. The description about fracture mechanism and failure mode detail have been added as shown in line 238-283.

  1. The English expression might be improved by a native English speaker.

Dear reviewer, thanks a lot. The English expression for this manuscript has been improved.

Reviewer 3 Report

Authors presented technically useful data, which will be meaningful only for engineers working in real field.

However, SEM measurement should be carried out to show dispersion and morphology of Glass Fibre in the composite.  

The title of this journal is “Materials” which means submitted manuscripts should have academic and scientific results and discussions for materials itself used in the study. 

However, all materials used in this study are commercialized products. This means there was no results and discussions for materials itself. 

Authors just presented the testing method and results for composite prepared using mass-produced commercially available products.

All presented data would be very useful for real field engineers, so I think this manuscript will attract more attention if it will be submitted to a mechanical engineering journal.

If authors want to publish their results in this journal, authors should explain the scientific and academic reasons for using E-glass fibre obtained from CA Composites Limited (Shang hai, China) 

and detail information about epoxy resin used. 

Mechanical properties of composite strongly depend on type of E-glass and epoxy resin used in the fabrication of the composite. 

Suggested results and discussions can not be useful and suitable for all kinds of the composites.

Without any discussion for materials itself, this manuscript could not get any attention from a material engineers and scientists.

No comment

Author Response

Dear editor,

Thanks for your letter and the thoughtful comments from the referees about our paper entitled “Influence of Woven Glass-Fibre Prepreg Orientation on Flexural Properties of a Sustainable Composite Honeycomb Sandwich Panel for Structural Applications” by A. L. Amir, M. R. Ishak, N. Yidris, M.Y.M. Zuhri, M. R. M. Asyraf and S. Z. S. Zakaria for consideration to be published in Materials Journal. We carefully analysed all the comments and these comments are very valuable and helpful for perfecting and modifying our manuscript, and also have important guiding significance for our research. Therefore, we carefully checked the manuscript and revised it according to each comment. Consequently, we feel that our manuscript is substantially strengthened. Revised portion are marked using yellow background in the revised manuscript. The detailed corrections in the paper and the responses to the reviewer’s comments are as the following list of revisions.

We look forward to your positive response. If you have any question about this paper, please don’t hesitate to let us know. We hope these revisions will make it more acceptable for publication. Thank you.

Sincerely yours,

Amir

Faculty of Engineering,

Universiti Putra Malaysia,

81310, UPM Serdang,

Selangor, Malaysia

Reviewer 3

  1. Authors presented technically useful data, which will be meaningful only for engineers working in real field. However, SEM measurement should be carried out to show dispersion and morphology of Glass Fibre in the composite.  

Dear reviewer, thanks for the constructive comment. Due to limited budget and time constrain, the SEM measurement can be carried out. However, it will become a gap for this paper to be fulfil in the future as discussed in conclusion.

  1. However, all materials used in this study are commercialized products. This means there was no results and discussions for materials itself. 

Authors just presented the testing method and results for composite prepared using mass-produced commercially available products.

All presented data would be very useful for real field engineers, so I think this manuscript will attract more attention if it will be submitted to a mechanical engineering journal.

If authors want to publish their results in this journal, authors should explain the scientific and academic reasons for using E-glass fibre obtained from CA Composites Limited (Shang hai, China) 

and detail information about epoxy resin used. 

Mechanical properties of composite strongly depend on type of E-glass and epoxy resin used in the fabrication of the composite. 

Suggested results and discussions can not be useful and suitable for all kinds of the composites.

Without any discussion for materials itself, this manuscript could not get any attention from a material engineers and scientists.

Dear reviewer, we have revised the manuscript as your comments in 287-306 line. Thanks a lot.

Reviewer 4 Report

The conclusions of the research are quite puzzling.

- The first one is contradictory (pages 11-12 – lines 364to367):

"All type of woven glass fibre prepreg orientation angle, α= 0Ëš, 45Ëš and 90Ëš facing almost the same failure mode due to minor difference in geometry and location of honeycomb cells like indentation failure, shear failure, wrinkling failure, buckling 366 failure, delamination failure, debonding failure and fibre breakage failure".

Besides the very poor English language in this sentence the conclusion is that all the types of woven glass fibre prepreg orientation angle, α= 0Ëš, 45Ëš and 90Ëš  facing almost the same failure mode.

If the failure modes are the same for all the three different orientations, why consider the rest of results? It is quite impossible that failure modes at 0 degrees and 45 degrees are the same. One must be subjected to the maximum tensile stress the second to the maximum shear stress. There is no way that either of the mentioned failure modes, e.g. bucking or delamination, would be the same in these two directions.

- The conclusions number 4 (page 12 – lines 374-376) is again senseless.

"The woven glass fibre prepreg orientation angle, α= 0Ëš, 45Ëš and 90Ëš have maximum stress and flexural modulus which directly depending upon the maximum load value obtained in three-point bending test".

Again, the poor English language might seclude the proper meaning of this conclusion. As I understand what is written, it does not matter which is the orientation of the fibres, the stress and the flexural modulus will be at their respective maximum values, depending on the maximum load value.

Since when the material's characteristics, the shear modulus, depend on the load value?

- General conclusions, page 12 – lines 393-394:

"This research will become the baseline for future works especially to be used as enhancement material in structures field".

I hope that authors meant the "baseline for THEIR future works, not in general. That would be a bit too presumptuous.

Further explanations are needed:

- Page 8 – lines 240-241:

"If compared the thickness of sandwich panel dimensions, the facesheet will become 240 the thinner dimension".

This sentence is senseless. It must be clarified what authors meant to say.

- Page 9 – lines 270-273:

"This tendency of the fibres to reorient towards the loading direction in a woven composite compared to α= 0Ëš and 90Ëš which mostly the load taken by the greater and stronger arrangement of reinforcing fibre/matrix [34]".

The second part of this sentence (after "which") is totally senseless. It must be rewritten to make sense.

- Page 10 – lines 304-306:

"However, Table 2 has shown that, there are different occur on α= 90Ëš suspected due to lay-up orientation angle during fabrication process".

This sentence makes no sense. Please, clarify the context.

There are many points where the authors are mixing singular and plural (noun-verb combination).
There are also several cases when they do not make difference between the noun (difference) and adverb (different), etc.

In general, the English language in this article is unacceptable. Authors should consult the English language lector to rewrite the WHOLE text.

The role of a scientific editor is not to correct the language to be able to understand what the authors have meant to say.

The scanned pages with marked errors and suggested corrections (where it was possible) are enclosed.

This article might have a scientific value, though there are some contradictory and unacceptable conclusions. The problem is that the English language is so poor that there are parts that I could not understand what was meant to say.

The text should undergo thorough review and reconstruction.

The role of a scientific editor is not to correct the language to be able to understand what the authors have meant to say.

The scanned pages with marked errors and suggested corrections (where it was possible) are enclosed.

Author Response

Dear editor,

Thanks for your letter and the thoughtful comments from the referees about our paper entitled “Influence of Woven Glass-Fibre Prepreg Orientation on Flexural Properties of a Sustainable Composite Honeycomb Sandwich Panel for Structural Applications” by A. L. Amir, M. R. Ishak, N. Yidris, M.Y.M. Zuhri, M. R. M. Asyraf and S. Z. S. Zakaria for consideration to be published in Materials Journal. We carefully analysed all the comments and these comments are very valuable and helpful for perfecting and modifying our manuscript, and also have important guiding significance for our research. Therefore, we carefully checked the manuscript and revised it according to each comment. Consequently, we feel that our manuscript is substantially strengthened. Revised portion are marked using yellow background in the revised manuscript. The detailed corrections in the paper and the responses to the reviewer’s comments are as the following list of revisions.

We look forward to your positive response. If you have any question about this paper, please don’t hesitate to let us know. We hope these revisions will make it more acceptable for publication. Thank you.

Sincerely yours,

Amir

Faculty of Engineering,

Universiti Putra Malaysia,

81310, UPM Serdang,

Selangor, Malaysia

Reviewer 1

  1. The conclusions of the research are quite puzzling.

- The first one is contradictory (pages 11-12 – lines 364to367):

"All type of woven glass fibre prepreg orientation angle, α= 0Ëš, 45Ëš and 90Ëš facing almost the same failure mode due to minor difference in geometry and location of honeycomb cells like indentation failure, shear failure, wrinkling failure, buckling 366 failure, delamination failure, debonding failure and fibre breakage failure".

Besides the very poor English language in this sentence the conclusion is that all the types of woven glass fibre prepreg orientation angle, α= 0Ëš, 45Ëš and 90Ëš  facing almost the same failure mode.

If the failure modes are the same for all the three different orientations, why consider the rest of results? It is quite impossible that failure modes at 0 degrees and 45 degrees are the same. One must be subjected to the maximum tensile stress the second to the maximum shear stress. There is no way that either of the mentioned failure modes, e.g. bucking or delamination, would be the same in these two directions.

Dear reviewer, thanks for the comment. The failure mode section has been revise with detail failure in Table form and discussion on it. Thank you.

  1. The conclusions number 4 (page 12 – lines 374-376) is again senseless.

"The woven glass fibre prepreg orientation angle, α= 0Ëš, 45Ëš and 90Ëš have maximum stress and flexural modulus which directly depending upon the maximum load value obtained in three-point bending test".

Again, the poor English language might seclude the proper meaning of this conclusion. As I understand what is written, it does not matter which is the orientation of the fibres, the stress and the flexural modulus will be at their respective maximum values, depending on the maximum load value.

Since when the material's characteristics, the shear modulus, depend on the load value?

Dear reviewer, thanks a lot for the comment. The sentence has been revised.

  1. General conclusions, page 12 – lines 393-394:

"This research will become the baseline for future works especially to be used as enhancement material in structures field".

I hope that authors meant the "baseline for THEIR future works, not in general. That would be a bit too presumptuous.

Dear reviewer, thanks a lot for the comment. The conclusion section has been revised.

  1. Further explanations are needed:

- Page 8 – lines 240-241:

"If compared the thickness of sandwich panel dimensions, the facesheet will become 240 the thinner dimension".

This sentence is senseless. It must be clarified what authors meant to say.

Dear reviewer, thanks a lot for the comment. The sentences has been revised.

  1. Page 9 – lines 270-273:

"This tendency of the fibres to reorient towards the loading direction in a woven composite compared to α= 0Ëš and 90Ëš which mostly the load taken by the greater and stronger arrangement of reinforcing fibre/matrix [34]".

The second part of this sentence (after "which") is totally senseless. It must be rewritten to make sense.

Dear reviewer, thanks a lot for the comment. The sentences has been revised

  1. Page 10 – lines 304-306:

"However, Table 2 has shown that, there are different occur on α= 90Ëš suspected due to lay-up orientation angle during fabrication process".

This sentence makes no sense. Please, clarify the context.

Dear reviewer, thanks a lot for the comment. The sentences has been revised

  1. There are many points where the authors are mixing singular and plural (noun-verb combination).
    There are also several cases when they do not make difference between the noun (difference) and adverb (different), etc.

In general, the English language in this article is unacceptable. Authors should consult the English language lector to rewrite the WHOLE text.

The role of a scientific editor is not to correct the language to be able to understand what the authors have meant to say.

The scanned pages with marked errors and suggested corrections (where it was possible) are enclosed.

Dear reviewer, we have revised the manuscript as your comment. Thank you very much.

Round 2

Reviewer 1 Report

Accept.

Author Response

Dear Reviewer,

Thanks a lot for your comment and support. Thank you.

Sincerely yours,

Amir

Faculty of Engineering,

Universiti Putra Malaysia,

81310, UPM Serdang,

Selangor, Malaysia

Reviewer 4 Report

Please, consult the scanned pages with proposed corrections.

The paper is now much improved, however, there are a few items that require corrections.

The scanned pages of the manuscript with marked errors and proposed corrections are enclosed.

The English language is improved, please pay attention to using singular and plural (verb/noun combination).

Author Response

Dear Reviewer,

Thanks a lot for the constructive comment.  We have revised the manuscript as per your comment. Thank you very much.

Sincerely yours,

Amir

Faculty of Engineering,

Universiti Putra Malaysia,

81310, UPM Serdang,

Selangor, Malaysia